# Mapping the immunogenic landscape of near-native HIV-1 envelope trimers in non-human primates

Christopher A. Cottrell[1,2,3], Jelle van Schooten[4], Charles A. Bowman[1], Meng Yuan[1], David Oyen[1], Mia Shin[1], Robert Morpurgo[4], Patricia van der Woude[4], Mariëlle van Breemen[4], Jonathan L. Torres[1], Raj Patel[1], Justin Gross[1], Leigh M. Sewall[1], Jeffrey Copps[1], Gabriel Ozorowski[1,2,3], Bartek Nogal[1,2,3], Devin Sok[2,3,5], Eva G. Rakasz[6], Celia Labranche[7], Vladimir Vigdorovich[8], Scott Christley[9], Diane G. Carnathan[3,10], D. Noah Sather[8], David Montefiori[7,11], Guido Silvestri[3,10], Dennis R. Burton[2,3,5,12], John P. Moore[13], Ian A. Wilson[1,2,3,14], Rogier W. Sanders[4,13]*, Andrew B. Ward[1,2,3]*, Marit J. van Gils[4]*

1 Department of Integrative Structural and Computational Biology, The Scripps Research Institute, La Jolla, California, United States of America, 2 IAVI Neutralizing Antibody Center, The Scripps Research Institute, La Jolla, California, United States of America, 3 Consortium for HIV/AIDS Vaccine Development, The Scripps Research Institute, California, United States of America, 4 Department of Medical Microbiology, Amsterdam UMC, University of Amsterdam, Amsterdam, The Netherlands, 5 Department of Immunology and Microbiology, The Scripps Research Institute, La Jolla, California, United States of America, 6 Wisconsin National Primate Research Center, University of Wisconsin, Madison, Wisconsin, United States of America, 7 Department of Surgery, Duke University Medical Center, Durham, North Carolina, United States of America, 8 Center for Global Infectious Disease Research, Seattle Children's Research Institute, Seattle, Washington, United States of America, 9 Department of Population and Data Sciences, UT Southwestern Medical Center, Dallas, Texas, United States of America, 10 Yerkes National Primate Research Center, Emory University, Atlanta, Georgia, United States of America, 11 Duke Human Vaccine Institute, Duke University Medical Center, Durham, North Carolina, United States of America, 12 The Ragon Institute of Massachusetts General Hospital, Massachusetts Institute of Technology and Harvard University, Cambridge, Massachusetts, United States of America, 13 Department of Microbiology and Immunology, Weill Medical College of Cornell University, New York, New York, United States of America, 14 The Skaggs Institute for Chemical Biology, The Scripps Research Institute, La Jolla, California, United States of America

* r.w.sanders@amsterdamumc.nl (RWS); andrew@scripps.edu (ABW); m.j.vangils@amsterdamumc.nl (MJVG)

**Data Availability Statement:** Germline database is available at http://ward.scripps.edu/gld/. The accession numbers for Env-specific BCR sequences are DDBJ/ENA/GenBank: MT002976-

## Abstract

The induction of broad and potent immunity by vaccines is the key focus of research efforts aimed at protecting against HIV-1 infection. Soluble native-like HIV-1 envelope glycoproteins have shown promise as vaccine candidates as they can induce potent autologous neutralizing responses in rabbits and non-human primates. In this study, monoclonal antibodies were isolated and characterized from rhesus macaques immunized with the BG505 SOSIP.664 trimer to better understand vaccine-induced antibody responses. Our studies reveal a diverse landscape of antibodies recognizing immunodominant strain-specific epitopes and non-neutralizing neo-epitopes. Additionally, we isolated a subset of mAbs against an epitope cluster at the gp120-gp41 interface that recognize the highly conserved fusion peptide and the glycan at position 88 and have characteristics akin to several human-derived broadly neutralizing antibodies.

MT002992 and MT008262-MT008328. RM IgM BCR sequences are available under BioProject ID: PRJNA604386. Atomic coordinates and structure factors of the reported crystal structure have been deposited in the Protein Data Bank (PDB: 6VOS, 6VOR, 6VSR). Cryo-EM reconstructions have been deposited in the Electron Microscopy Data Bank (EMDB: EMD-21246, EMD-21257, EMD-21232), and in the Protein Data Bank (PDB: 6VN0, 6VO1, 6VLR). The accession numbers for the negative stain 3D EM reconstructions are Electron Microscopy DataBank: EMD-21053-21059, EMD-21061-21062, EMD-21064-21066, EMD-21075, EMD-21076, EMD-21077, EMD-21078, EMD-21080, EMD-21081, EMD-21082-21093, EMD-21272-21278.

**Funding:** This work was supported by the HIV Vaccine Research and Design (HIVRAD) program (P01 AI110657) (A.B.W., R.W.S, J.P.M., and I.A. W.), NIH CHAVI-ID (UM1 AI100663) and CHAVD (UM1 AI44462) awards (A.B.W., I.A.W. and D.R. B.), NIH R01 AI13082 (J.P.M.), the International AIDS Vaccine Initiative Neutralizing Antibody Center, the Bill and Melinda Gates Foundation CAVD (OPP1115782, OPP1132237, OPP1084519, OPP119635), and the European Union's Horizon 2020 research and innovation program under grant agreement no. 681137 (R.W.S.). C.A.C. is supported by the NIH F31 Ruth L. Kirschstein Predoctoral Award AI131873 and by the Achievement Rewards for College Scientists Foundation. R.W.S. is a recipient of a Vici fellowship from the Netherlands Organization for Scientific Research (NWO). G.O. and M.J.v.G. are supported by amfAR Mathilde Krim Fellowships in Basic Biomedical Research grant numbers 109718-63-RKNT and 109514-61-RKVA, respectively. J.v.S. is a recipient of a 2017 AMC Ph. D. Scholarship. Computational analyses of EM data were performed using shared instrumentation funded by NIH grant S10OD021634. Naïve B-cell sorting and NGS sequencing was conducted as part of the Technologies Across Scale graduate course with support from Skaggs Graduate School of Chemical and Biological Sciences at Scripps Research. Use of the Stanford Synchrotron Radiation Lightsource, SLAC National Accelerator Laboratory, is supported by the U.S. Department of Energy, Office of Science, Office of Basic Energy Sciences under Contract No. DE-AC02-76SF00515. The SSRL Structural Molecular Biology Program is supported by the DOE Office of Biological and Environmental Research, and by the National Institutes of Health, National Institute of General Medical Sciences (including P41GM103393). The contents of this publication are solely the responsibility of the authors and do not necessarily

## Author summary

Major efforts are currently directed towards developing vaccine strategies to successfully elicit broadly neutralizing antibodies (bNAbs) against HIV-1. However, how to achieve this goal remains a critical problem. Soluble native-like HIV-1 envelope glycoproteins (Env) have shown promise as vaccine candidates in pre-clinical studies. Here, the antibody response against the BG505 SOSIP.664 Env trimer was studied through the isolation of monoclonal antibodies (mAbs) in rhesus macaques (RMs) after immunization. Overall, a diverse landscape of mAbs was elicited that target the Env trimer in highly similar ways in two animals. By mapping the target epitopes of all of the mAbs isolated from the immunized RMs, rather than focusing only on the neutralizing mAbs, we were able to identify several non-neutralizing and potentially immunodominant epitopes that would ideally be eliminated in future immunization studies. In addition, we identified neutralizing mAbs that recognize the highly conserved fusion peptide in a similar way as human broadly neutralizing antibodies. These insights can be further used to develop immunogens and immunization strategies able to induce bNAb-like responses that can protect against infection.

## Introduction

HIV-1 continues to cause significant morbidity and mortality around the world with an estimated 1.7 million new infections in 2018 [1], which emphasizes the need for an effective prophylactic vaccine. The HIV-1 envelope glycoprotein (Env) is the sole target for neutralizing antibody (NAb) responses. Studies of infected patients have led to the isolation of NAbs against multiple different epitopes on the Env surface that are capable of both neutralizing most circulating strains and providing passive protection against repeated viral challenges in non-human primates (NHPs) [2–5]. Extensive research efforts, including structure-based engineering of Env immunogens, are currently directed towards developing vaccine strategies to successfully elicit broadly neutralizing antibodies (bNAbs) against specific Env epitopes [6–12]. The development and structural determination of soluble native-like Env trimer mimics, particularly ones based on the SOSIP technology, has provided a platform for structure-based immunogen design [13–16]. NAbs induced by SOSIP trimers in NHPs can protect against challenge with an autologous Simian-Human Immunodeficiency virus (SHIV) [17,18]. However, as NAbs with the required breadth of activity have not yet been induced in trimer-immunized animals, improvements to current vaccine design and delivery strategies are clearly needed.

Characterizing the antibody response to SOSIP trimers may provide useful information for guiding immunogen design. Initial analyses of rabbits immunized with BG505 SOSIP.664 trimers, including studies of isolated monoclonal antibodies (mAbs), showed that autologous NAbs targeted a large hole in the glycan shield of the BG505 virus caused by the absence of glycosylation sites at positions 241 and 289 [19,20]. The 241 glycan is highly conserved (97%) among HIV-1 strains and while the 289 glycan is less conserved, it is still present in 79% of viruses. The required absence of typically conserved glycans explains why the NAbs isolated from the trimer-immunized rabbits lack breadth [19]. Later studies involving BG505 trimer-immunized rabbits, guinea pigs, and NHPs have identified additional narrow-specificity neutralizing serum responses that recognize epitopes in the C3/V4, C3/V5, and V1 regions, with mAbs isolated from guinea pigs targeting the C3/V4 epitope [17,20–23].

represent the official views of NIGMS, NIAID, or NIH.The funders had no role in study design, data collection and analysis, decision to publish, or preparation of the manuscript.

**Competing interests:** The authors have declared that no competing interests exist.

Here, we describe a detailed characterization of neutralizing and non-neutralizing mAbs isolated from two rhesus macaques (RMs) previously immunized with the BG505 SOSIP.664 trimer [24]. We identified multiple mAbs targeting the 289-glycan hole on the BG505 SOSIP.664 trimer or a neo-epitope cluster at the base of the trimer. The most potent NAb isolated targeted the gp120/gp41 interface at an epitope that significantly overlaps with the epitope of human bNAb VRC34 [25]. Insights from the induction of these NAbs through vaccination can be further used to develop immunogens and immunization strategies to induce cross-reactive antibody responses.

## Results

### Indian origin rhesus macaque BCR germline database

The majority of bNAbs isolated from HIV-infected patients have exceedingly high levels of somatic hypermutation (SHM) [5,26]. Accurately measuring levels of SHM elicited during immunization experiments is a critical component to ensuring the elicited antibodies are acquiring the level of mutations associated with neutralization breadth. To accurately measure the extent of SHM, the mAb sequences that we obtained from BG505 SOSIP.664 trimer-immunized RMs required comparison to a germline B-cell receptor (BCR) reference database. The IMGT reference database for RMs is incomplete and contains a mixture of genes/alleles from both Chinese and Indian origin animals. Given the high levels of genetic diversity in the Immunoglobulin (Ig) loci among RMs from different origins [27] and the general use of Indian origin RMs in most HIV-1 immunization experiments conducted in the United States, we constructed a germline database containing gene/alleles from only Indian origin RMs genomic data. The resulting database contained 189 IGHV, 70 IGHD, 9 IGHJ, 188 IGKV, 5 IGKJ, 147 IGLV, and 13 IGLJ genes/alleles (S1 Table).

Recent advances in BCR repertoire sequencing and analysis have enabled the use of next-generation sequencing (NGS) datasets for inferring novel genes/alleles [27–29]. Using the database described above as an initial database, we performed IgDiscover analysis [27] on IgM BCR sequences derived from five Indian origin RMs. Additionally, we performed IgDiscover analysis on previously obtained Indian origin RM BCR NGS datasets that were downloaded from the NCBI sequence read archive [27,30] or obtained directly from the study authors [31]. We added 113 IGHV, 18 IGKV, and 18 IGLV genes/alleles to our germline database (http://ward.scripps.edu/gld/) resulting in a total of 302 IGHV, 206 IGKV, and 165 IGLV genes/alleles.

### Antigen-specific mAbs isolated from BG505 SOSIP.664 trimer-immunized RMs

To better understand the immunogenicity of the BG505 SOSIP.664 trimers in previously immunized Indian origin RMs [24] we selected the two animals (rh1987 and rh2011) with the highest serum neutralization activity against the autologous BG505.T332N pseudovirus ($ID_{50}$ of 292 and 266, respectively at 2 weeks post sixth immunization) [24] for in-depth mAb analysis. Peripheral blood mononuclear cells (PBMCs) from the following time points were selected for BG505 SOSIP.664 trimer-specific IgG-positive single memory B-cell sorting: (i) two weeks prior to the fourth immunization (week 22), (ii) 1 week after the fourth immunization (week 25) and (iii) 1 week after the sixth immunization (week 53) (Fig 1A). In total, 25 and 17 mAbs were cloned from RMs rh1987 and rh2011, respectively (Fig 1B).

The BG505-specific mAb sequences were analysed using our germline database and shown to be evenly distributed between kappa and lambda light chains (KC and LC) usage (Fig 1B).

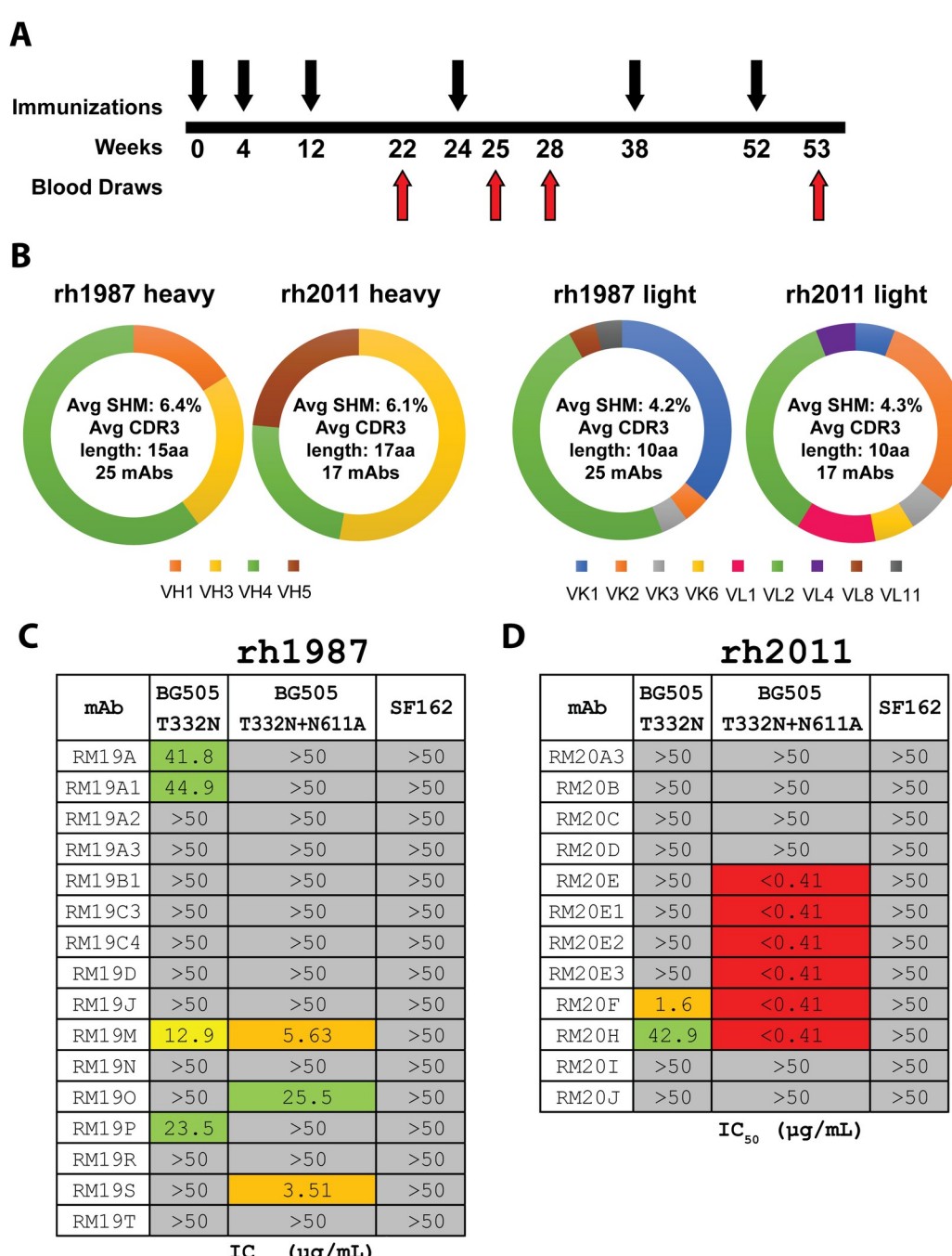

**Fig 1. MAb isolation and characterization from BG505 SOSIP.664 trimer-immunized RMs.** (A) Simplified immunization scheme as described in Sanders et al. 2015 [24]. Black arrows indicate i.m. immunizations with 100 μg of BG505 SOSIP.664 with 75 units of ISCOMATRIX adjuvant. Red arrows indicate blood draws. (B) Heavy and light chain genetic characteristics for mAbs isolated from rh1987 and rh2011. (C) TZM-bl neutralization of BG505.T332N, BG505.T332N.N611A, and SF162 pseudoviruses for mAbs isolated from RM rh1987. (D) TZM-bl neutralization of BG505.T332N, BG505.T332N.N611A, and SF162 pseudoviruses for mAbs isolated from RM rh2011. Assay limit of detection was at an $IC_{50}$ of 50 μg/mL.

For animal rh1987, 11 KC and 14 LC mAbs were isolated. Their average heavy chain (HC) SHM (nucleotide level) was 6.4% (range: 2.1%-10.2%) with an average HC complementarity-determining region 3 (CDR-H3) length of 15 amino acids (aa) (range: 7–23) (S2 Table). The

rh1987 KC mAbs utilized HC variable genes from the IGHV3 and IGHV4 families and predominantly used KC V genes from the IGKV1 family (S2 Table). All of the rh1987 KC mAbs had a CDR-L3 length of 9 aa and their average KC SHM (nucleotide level) was 4.7% (range: 2.6%-6.0%) (S2 Table). A single clonal family with 4 members (RM19A) was detected among rh1987 KC mAbs with members isolated from both week 22 and week 25 samples (S2 Table). The rh1987 LC mAbs used HC V genes from the IGHV1, IGHV3 and IGHV4 families and LC V genes mainly from the IGLV2 gene family (S2 Table). The rh1987 LC mAbs had an average CDR-L3 length of 10 aa (range: 9–11) with an average LC SHM (nucleotide level) of 3.8% (range: 0.9%-10.6%) (S2 Table). Two clonal families (RM19B [2 members] and RM19C [4 members]) were identified among the rh1987 LC mAbs isolated from weeks 22 and 25 (S2 Table).

For animal rh2011, 8 KC and 9 LC mAbs were isolated. Their average HC SHM rate (nucleotide level) was 6.1% (range: 3.0%-9.1%) and they had an average CDR-H3 length of 17 aa (range 10–20) (S2 Table). Half of the rh2011 KC mAbs belonged to the RM20E clonal family, isolated from the week 53 sample. The RM20E clonal family utilized the HC V gene IGHV5-ABI*01_S2502 and the KC V gene LJI.Rh_IGKV2.71 (S2 Table). Overall, the rh2011 KC mAbs had an average KC SHM rate (nucleotide level) of 4.1% (range: 3.0%-5.3%) and a CDR-L3 length of 9 aa (S2 Table). Among the rh2011 LC mAbs, two clonal families (RM20A [4 members] and RM20B [2 members]) were isolated from weeks 22 and 25 samples (S2 Table). Overall, the rh2011 mAbs had an average LC SHM rate (nucleotide level) of 4.4% (range: 2.1%-6.4%) and an average CDR L3 length of 10.6 aa (range: 9–11) (S2 Table).

## BG505-specific mAbs recognize multiple Env regions

All 42 mAbs bound to the BG505 SOSIP.664 trimer in ELISA, but only 11 of the 25 mAbs from rh1987 and 9 of the 17 from rh2011 bound the corresponding gp120 monomer (S1 Fig and S3 Table). The mAbs were tested for neutralization activity against the autologous BG505 clade A Tier 2 virus, its glycan-611 knockout variant (N611A), and the heterologous SF162 clade B Tier 1A virus. Only a few mAbs, 4 from rh1987 and 2 from rh2011, neutralized the BG505.T332N pseudovirus but one of them, RM20F from rh2011, did so potently (Fig 1C and S3 Table). Two mAbs from rh1987 and 4 from rh2011 were able to potently neutralize the N611A-variant despite lacking activity against the autologous BG505.T332N pseudovirus (Fig 1C and 1D). None of the 42 mAbs neutralized the easy-to-neutralize heterologous SF162 virus (Fig 1C and 1D).

## EM-based epitope mapping revealed mAbs isolated from both animals target 4 distinct, but somewhat overlapping epitopes

We used low resolution, negative stain, single particle electron microscopy (EM) to visualize where a representative subset of the isolated mAbs bound on the surface of the BG505 SOSIP trimer. The majority (55%) of mAbs isolated were non-neutralizing antibodies that bound to the base of the BG505 SOSIP trimer (Fig 2A, S2 Fig, S2 Table, and S3 Table) at a neo-epitope cluster that is occluded by the viral membrane on HIV-1 virions. Fabs bound to the base of the soluble trimer via multiple angles of approach and utilized a variety of heavy and light chain genes/alleles to do so (Fig 2A and S2 Table). The extent of SHM in the base-targeting mAbs ranged from 2–10% in the HC and 1–11% in the light chain (S2 Table). Previous studies examining the polyclonal antibody responses elicited by the BG505 SOSIP trimers in rabbits and RMs have shown that epitopes at the base of the soluble trimer were targeted in every single animal analyzed [21,23,32]. Taken together with our new data, it is clear that the base of SOSIP trimers contain an immunodominant non-neutralizing neo-epitope cluster that is easily targeted by a variety of precursor BCRs in different species.

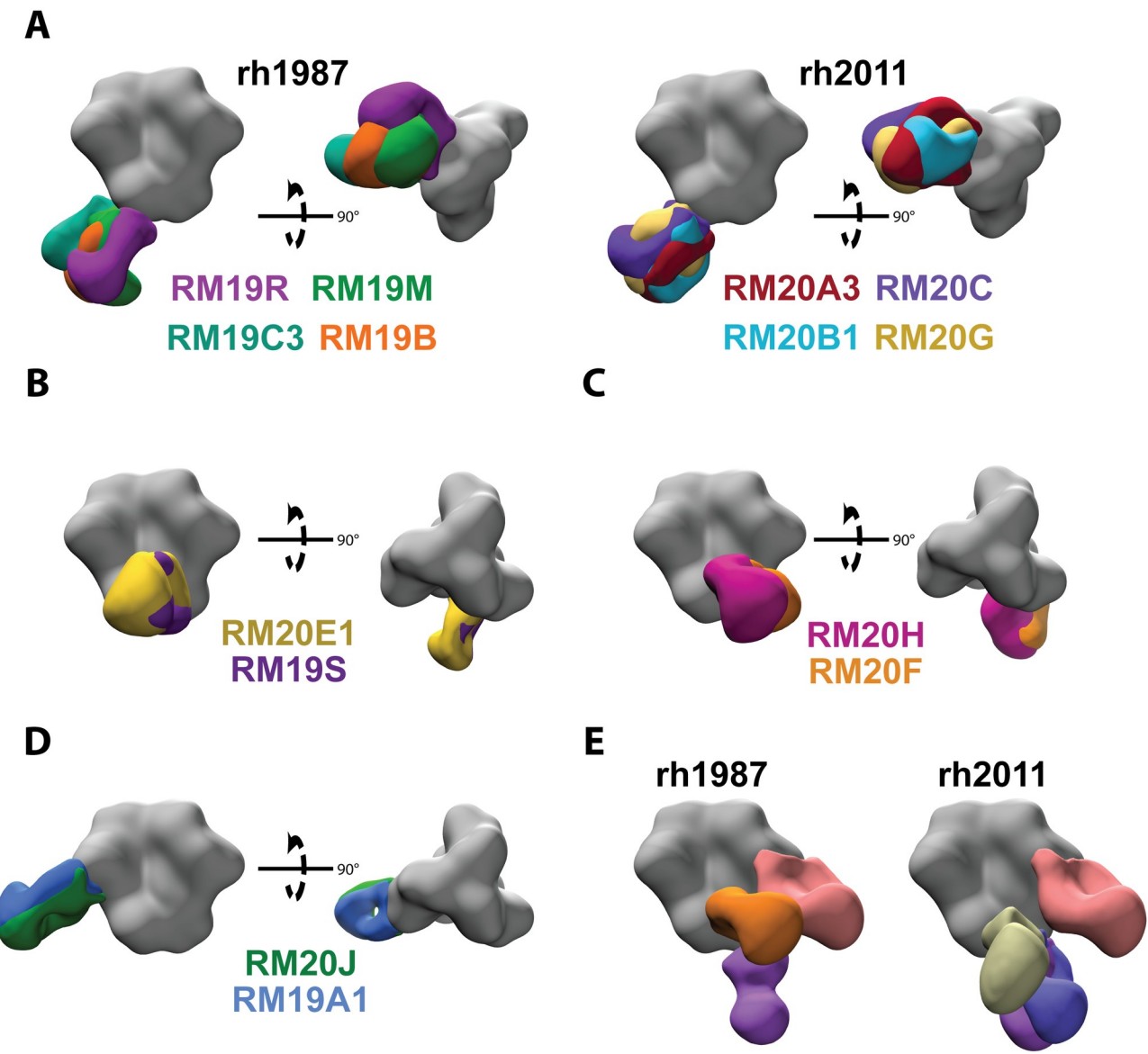

**Fig 2. Epitope mapping by negative stain electron microscopy.** (A) Representative base-targeting mAbs for animals rh1987 and rh2011. (B) Overlapping gp120/gp41 interface epitope targeted by mAbs from both animals. (C) Overlapping FP epitope targeted by mAbs from both animals. (D) Overlapping N289-glycan hole epitope targeted by mAbs from both animals. (E) EMPEM analysis for wk28 IgG from rh2011 and rh1987. All structural figures were generated with UCSF Chimera [62]. EM density maps were segmented with the Segger extension in UCSF Chimera [62,63].

A subset of mAbs from both animals bound to an epitope near the N611 glycan (Figs 2B and S2). These mAbs were not capable of neutralizing the autologous BG505.T332N pseudovirus but neutralized the BG505 N611A variant (Fig 1C and 1D). Two additional mAbs isolated from rh2011 (RM20F and RM20H) bound to an epitope near the fusion peptide (FP) and were capable of neutralizing both the autologous and N611A BG505 pseudoviruses, but the latter virus more potently (Figs 2C and S2). MAbs from both animals targeted the 289-glycan hole epitope on BG505, with some, isolated from rh1987, showing weak neutralization of the autologous BG505.T332N pseudovirus (Figs 1C and 2D, S3 Table). Multiple germline genes/alleles were used to target the same 289-glycan hole epitope (S2 Table).

To further assess the epitopes targeted following immunization with the BG505 SOSIP.664 trimer and verify that we isolated mAbs representative of the full serum antibody response, we performed electron microscopy polyclonal epitope mapping (EMPEM) [32] using week 28 serum. EMPEM revealed that similar epitopes were targeted in both animals and that the epitope assignments correlated well with the epitopes ascribed to mAbs generated by antigen-specific B-cell sorting (Figs 2E versus S2).

A previous analysis of purified serum IgGs from RMs rh1987 and rh2011 identified the C3/V5 epitope as a major target for neutralization activity [20]; however, none of the mAbs isolated here targeted the C3/V5 epitope nor were they detected by EMPEM. While low resolution, negative stain EM provides valuable information on where mAbs bind on the surface of HIV-1 Env, the molecular detail necessary to guide structure-based immunogen design requires high-resolution structural data obtained by cryoEM and x-ray crystallography. We therefore selected three Fabs (RM20J, RM20F, and RM20E1) that bound to different epitopes for high-resolution structural determination.

## MAb RM20J binds to the α2 helix of gp120 and exploits a hole in the glycan shield of BG505 at position 289

We solved a 2.3 Å crystal structure of unliganded RM20J Fab and a 3.9 Å cryoEM structure of RM20J Fab bound to the BG505 SOSIP Env trimer (Fig 3A and S3 Fig, S4 Table and S5 Table). Together these structures revealed the RM20J Fab binds to an epitope on a single gp120 protomer with 982 Å$^2$ of buried surface area (BSA). The CDR-H1 and CDR-H2 make contact with the C2 region of gp120 including residues N289 and T290 (Fig 3B). A glycan at position N289 would directly clash with both the CDR-H1 and CDR-H2 of RM20J (Fig 3B). CDR-L2 makes contact with the first N-acetyl glucosamine sugar of the N355 glycan (Fig 3C). Additional contacts are made to the α2 helix of gp120 by RM20J CDR-H3 and CDR-L2 (Fig 3C). When compared to 10A, a previously characterized 241/289 glycan hole targeting NAb isolated from a BG505 SOSIP.664-immunized rabbit [19,32], RM20J binds to an epitope biased more towards 289 and away from 241 in the 241/289 glycan hole, revealing subtle differences in the recognition of the epitope (Fig 3D and 3E). Despite binding to the BG505 SOSIP trimer with high affinity (S6 Table), RM20J was not able to neutralize the autologous BG505.T332N pseudovirus (Fig 1C). Although the hypervariable region of V4 was not resolved in the trimer structure, it lies directly above the RM20J epitope and contains two additional glycans (N406 and N411) that may affect RM20J binding. Comparisons between the glycosylation profiles of the BG505 viral Env and the SOSIP.664 trimer revealed differences in the glycoforms present at positions N355, N406, and N411 [33,34] with more complex glycans being found on the viral Env that could hinder the ability of RM20J to bind on the surface of the virus and, therefore, render it incapable of neutralization. Several of the mAbs isolated from rh1987, including the RM19A clonal family, bind to a similar epitope as RM20J (Fig 2D, S2 Fig, S2 Table) and either fail to neutralize the autologous BG505 virus or do so with weak potency (Fig 1C and 1D, S3 Table).

## MAb RM20F binds to a quaternary epitope at the gp120/gp41 interface that includes elements of the fusion peptide and the N88 glycan

For a more detailed view of the mode of RM20F recognition, we solved a 2.2 Å crystal structure of unliganded RM20F Fab and a 4.3 Å cryoEM structure of RM20F Fab bound to BG505 SOSIP trimer (Fig 4A, S3 Fig, S4 Table, S5 Table). RM20F recognizes an epitope spanning two gp41 protomers and a single gp120 protomer that has 1126 Å$^2$ of BSA at the interface. The RM20F LC contributes 22% of the paratope surface area (250 Å$^2$) and makes contact with the

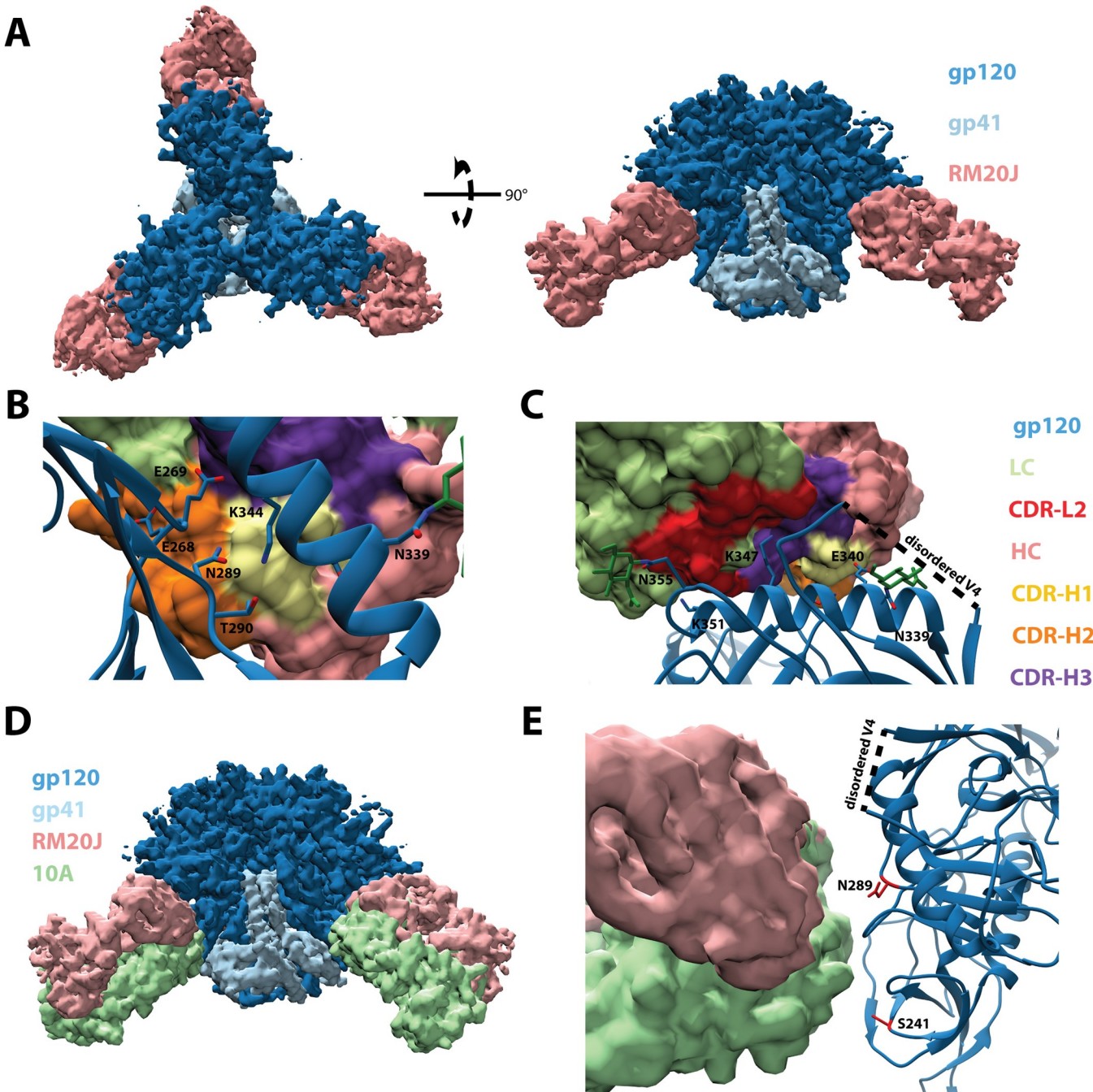

**Fig 3. RM20J binds to the N289 glycan hole region of BG505 SOSIP.v5.2.** (A) Segmented 3.9 Å cryoEM reconstruction of RM20J Fab (pink) in complex with BG505 SOSIP.v5.2 (gp120, dark blue; gp41, light blue). (B) and (C) Zoomed-in views of the epitope/paratope interaction between gp120 (blue, ribbon diagram) and RM20J Fab (surface representation). (D) and (E) Comparison of the RM20J epitope with that of the BG505 SOSIP.664 elicited rabbit neutralizing NAb 10A (PDB: 6DID) [19,32].

poorly conserved residues H85 (8.1% prevalence among global strains) and K229 (12.5% prevalence) in the C1 and C2 regions of gp120 respectively (Fig 4B). The RM20F HC contributes the remaining 78% of the paratope surface area (876 Å$^2$) and uses its 20 residue CDR-H3 to wedge between the FP of the primary gp41 protomer and the HR2 helix of the adjacent gp41

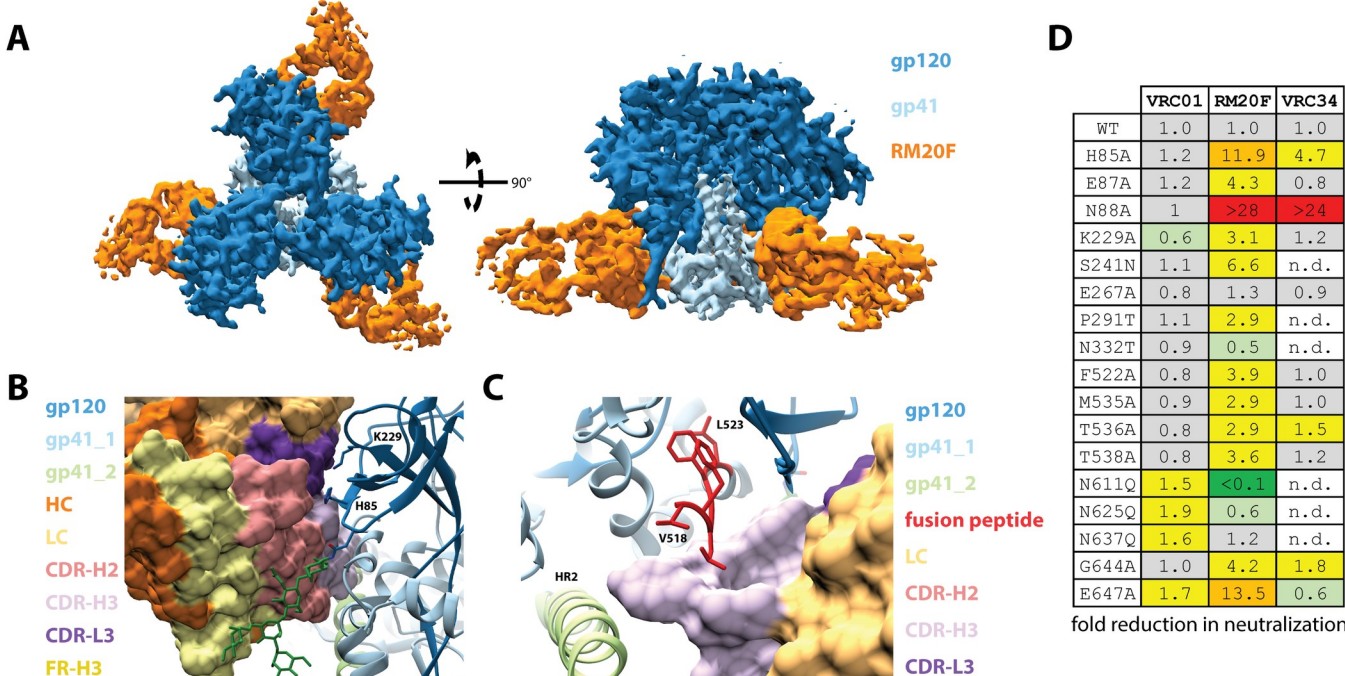

**Fig 4. RM20F binds to a quaternary epitope at the gp120/gp41 interface of BG505 SOSIP.v4.1.** (A) Segmented 4.3 Å cryoEM reconstruction of RM20F Fab (orange) in complex with BG505 SOSIP (gp120, dark blue; gp41, light blue). (B) and (C) Zoomed-in views of the epitope/paratope interaction between BG505 SOSIP (gp120, dark blue; gp41s, light blue and light green; ribbon diagram) and RM20F Fab (surface representation). (D) Neutralization data for BG505 mutant pseudoviruses against VRC01, RM20F, and VRC34.

protomer (Fig 4C). Additional contacts with the fusion peptide proximal region (FPPR) of the primary gp41 protomer are made by residues at the tip of CDR-H2 (Fig 4B). The N88 glycan accounts for 18% (198 Å$^2$) of the epitope BSA and makes contact with the CDR-H2 and FR-H3 regions of RM20F (Fig 4B). The lack of connecting density, even at lower contour, between RM20F and the glycans at N611 and N637 suggests these glycans do not substantially contribute to the epitope. Epitope mapping using BG505.T332N mutant pseudoviruses showed that knocking out the N611 glycan (N611Q mutant) substantially enhanced neutralization by RM20F, while knocking out the N637 glycan (N637Q mutant) had no effect. (Fig 4D). Other virus mutants revealed that neutralization by RM20F was sensitive to various sequence changes within the epitope, particularly at residues H85 and E647 (84.3% prevalence) and N88 (N88 glycan knock out) (Fig 4D). The N88 glycan knock out and the H85A mutation (to a lesser extent) significantly reduced neutralization activity of the FP-targeting bNAb VRC34, but no effect on the CD4 binding site targeting bNAb VRC01 (Fig 4D). Introducing the 241 or 289 glycans (S241N and P291T, respectively) modestly reduced the neutralization activity of RM20F (Fig 4D). In comparison to the FP-targeting bNAbs VRC34 and ACS202, RM20F lacked neutralization breadth when tested against a panel of multiple heterologous viruses (S7 Table) likely due to the dependency on poorly conserved residues, such as 85 and 229.

## MAb RM20E1 binds to the fusion peptide and makes contact with two adjacent protomers

We solved a 2.3 Å crystal structure of the unliganded RM20E1 Fab and 4.4 Å cryoEM structure of RM20E1 Fab bound to BG505 SOSIP trimer and Fab PGT122 (Fig 5A, S3 Fig, S4 Table, S5

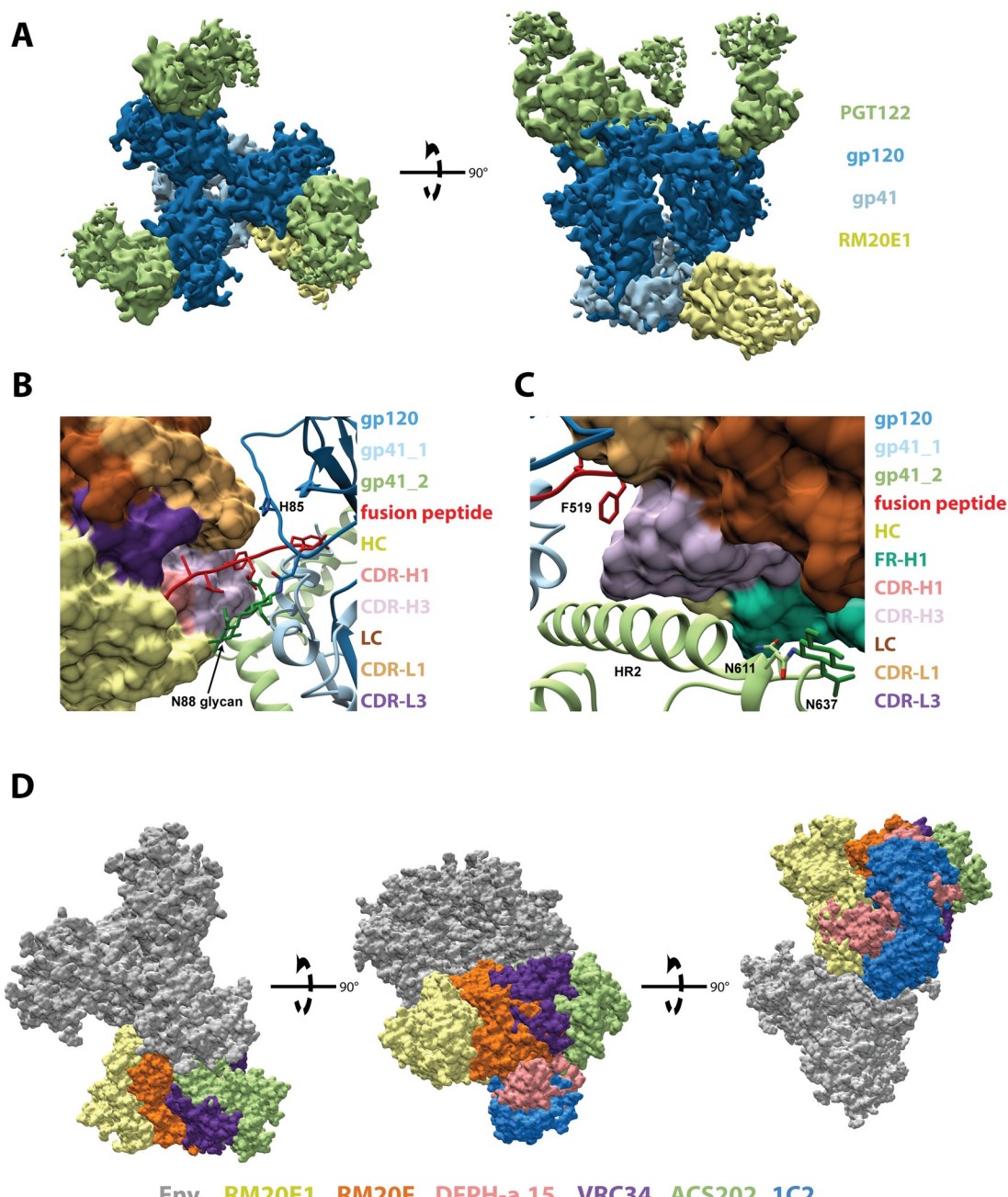

**Fig 5. RM20E1 binds to the fusion peptide of BG505 SOSIP.v5.2.** (A) Segmented 4.2 Å cryoEM reconstruction of RM20E1 Fab (yellow) and PGT122 Fab (light green) in complex with BG505 SOSIP (gp120, dark blue; gp41, light blue). (B) and (C) Zoomed-in views of the epitope/paratope interaction between BG505 SOSIP (gp120, dark blue; gp41s, light blue and light green; ribbon diagram) and RM20E1 Fab (surface representation). (D) Comparison of RM20E1 and RM20F epitopes to those of FP targeting bNAbs VRC34, ACS202, and DFPH-a.15 and the gp120/gp41 interface bNAb 1C2 [9,25,35–37].

Table). RM20E1 binds to an epitope composed of one gp120 and two gp41 protomers with 1178 Å$^2$ of BSA. Residues 515 to 520 of the FP in the primary gp41 protomer are sandwiched between CDR-H3, CDR-L1, and CDR-L3 of RM20E1 (Fig 5B). CDR-H3 and FR-H1 make contact with HR2 in the adjacent gp41 protomer (Fig 5C). Additionally, the FR-H1 makes

contact near the N611-glycan site in the adjacent gp41 protomer, but we observed no connecting density that could be attributed to the N611-glycan itself (Fig 5C). RM20E1 avoids the N88 glycan but does interact with residues in the C1 region of gp120, including H85, via its CDR-L1 (Fig 5B). Despite recognition of the conserved FP, RM20E1 did not neutralize the autologous BG505.T332N pseudovirus. The antibody did however potently neutralize the N611A glycan KO BG505 pseudovirus (Fig 1C) suggesting the epitope is shielded by the N611 glycan. The epitopes of the FP-targeting bNAbs VRC34, ACS202, and DFPH-a.15 overlap to a large extent with the epitopes of RM20E1 and RM20F (Fig 5D), with DFPH-a.15 and VRC34 also neutralizing more potently in the absence of the N611 glycan [9,25,35,36]. The RM20E1-bound FP conformation is similar to the FP conformation when bound by the bNAb VRC34 (S4 Fig); however, the inability of RM20E1 to accommodate the N611 glycan likely results in the lack of neutralization of the wild-type virus.

## Discussion

A major goal of HIV-1 vaccine research is to elicit bNAbs able to neutralize the large diversity of circulating HIV-1 strains in humans. However, how to achieve this goal remains a critical problem. Native-like Env trimers are an important design platform for engineering immunogens for bNAb induction [7–13,37,38]. BG505 SOSIP trimers were able to induce responses in immunized RMs that potently neutralized the autologous Tier 2 virus [17,24]. When present at sufficient titers, those NAbs protected against BG505-SHIV challenge [18]. Evaluating SOSIP trimers in RMs can yield valuable information because of the close genetic relationship between RMs and humans. Here, we isolated mAbs from two BG505 SOSIP.664 trimer-immunized RMs to better understand how the immune system recognizes the trimers and the epitopes associated with the potent, but limited, HIV-1 Tier 2 neutralization.

By mapping the epitopes of all of the mAbs isolated from the immunized RMs, rather than focusing only on NAbs, we were able to identify several non-neutralizing and potentially immunodominant epitopes that would ideally be eliminated in future immunization studies. As shown previously with mAbs from rabbits immunized with BG505 SOSIP.664 trimers [19], the lack of glycans at positions 241 and 289 in BG505 creates a large glycan hole which is targeted by mAbs from both RMs. The mAbs isolated from RMs that target the 241/289 glycan hole are more biased towards the 289-site compared to the previously characterized rabbit mAbs. This difference may be attributed to the underlying differences in BCR repertoires between the two animal models. In addition to the lack of glycans due to missing sequons that encode for glycosylation, the recombinant BG505 SOSIP.664 trimer may also contain missing glycans even when the correct sequon is present as previously observed by mass spectrometry studies of glycopeptides [33,39]. Our study identified gp120/gp41 interface antibodies whose neutralization was enhanced in the absence of the N611 glycan, suggesting that the BG505 SOSIP.664 trimer may have sub-stoichiometric glycan occupancy in gp41 at this position, creating an unexpected but immunogenic glycan hole. The elicitation of FP targeting mAbs in RMs with the BG505 SOSIP.664 trimer provides evidence that the FP bNAb epitope is accessible and immunogenic on soluble Env trimer immunogens. Recent studies in mice, guinea pigs and RMs using synthetically produced HIV fusion peptides covalently attached to carrier proteins as priming immunogens followed by boosts with soluble Env trimer immunogens have also elicited FP specific antibodies including some mAbs with neutralization breadth [9,40,41]. However, the majority of the animals immunized in these studies do not develop neutralization breadth and instead develop potent neutralization against the BG505 pseudovirus with the N611 glycan KO [9,40,41]. Given the consistency across studies and animal models in eliciting potent NAbs that target the FP epitope and require the absence of the N611 glycan,

investing in strategies to quantify and enhance the N611 glycan occupancy in soluble Env trimer immunogens, particularly for boost immunogens, may improve the neutralization breadth elicited by FP-targeting immunization protocols.

The sorting probe used to isolate BG505 Env-specific B-cells was a C-terminally biotinylated BG505 SOSIP.664 trimer bound to a fluorescent streptavidin tetramer. Steric constraints between the base of the trimer and the streptavidin tetramer likely resulted in a lower recovery of base epitope-specific B-cells. Despite this potential selection bias, 55% of the mAbs isolated from the two immunized RMs bound to the base of the BG505 SOSIP.664 trimer, indicating the base of the soluble trimer is the major target for antibody responses during immunization. To reduce the immunogenicity of this epitope, glycans can be introduced to shield this site or the native-like trimers could be constructed onto scaffolds or particles [42–45].

We were unable to construct individual germline BCR databases from RMs rh1987 and rh2011 to precisely determine the SHM and gene/alleles usage as additional PBMC samples were no longer available. Instead, we constructed a germline database containing BCR gene/alleles from multiple Indian origin RMs that allowed us to measure SHM levels in the mAbs we isolated from RMs. The new database provides web-based access (http://ward.scripps.edu/gld/) to a curated and highly annotated general resource for examining BCR gene/alleles from Indian origin RMs. Previous estimates of average SHM rates in mAb sequences from Env-immunized RMs were 8.9% and 6.1% for the HC and LC, respectively [46]. These apparently high levels of SHM following repeated immunization with the exact same soluble Env immunogens were likely due to missing germline gene/alleles from the database used to calculated SHM. Using the germline database reported in this study, we were able to assign vaccine-elicited antibody sequences to specific germline genes/alleles and determine levels of SHM much more accurately. The average levels of SHM reported in this study (6.3% for HC and 4.2% for LC) are comparable to the average levels of SHM reported in similar immunization studies where per animal germline BCR databases were inferred using IgDiscover [47,48]. Our germline database provides a resource for assigning germline genes/alleles and accurately calculating rates of SHM when inferring individual germline databases for each animal is logistically impractical.

In conclusion, in this study, neutralizing and non-neutralizing mAbs with distinctive epitopes were isolated and characterized in BG505 SOSIP.664-immunized RMs. We demonstrated that a polyclonal response was elicited in two different RMs that target the BG505 SOSIP.664 trimer in highly similar ways. While rabbit antibody responses are dominated by the base and 241-glycan hole epitopes, the RM mAbs target more diverse epitopes with mAbs also targeting the FP and gp120/gp41 interface. The mAbs characterized here also provide a valuable resource for epitope mapping and comparison to a wide array of BG505-based immunization experiments, including the recently initiated BG505 SOSIP.664 human clinical trial (NCT03699241). Finally, the FP-targeting mAbs in particular provide a structure-guided opportunity to modify BG505 SOSIP and other trimers to focus the antibody response on the FP-region, with the goal of eliciting bNAb-like antibodies.

## Methods

### Immunizations of rhesus macaques

Immunization samples used in this study were obtained from previously immunized RMs described in Sanders et al. 2015 [24]. Briefly, RMs were immunized intramuscularly (i.m.) with 100 μg of BG505 SOSIP.664 trimer formulated in 75 units of ISCOMATRIX given at week 0, 4, 12, 24, 38 and 52.

## Ethics statement

All immunizations and blood samplings were performed at the Wisconsin National Primate Research Center (NPRC) and approvals were obtained from the Wisconsin NPRC as described previously [24]. The study was carried out according to the NIH guidelines in compliance with IACUC regulations. The rhesus macaques were housed, immunized and bled in compliance with the Animal Welfare Act and other federal statutes and regulations relating to animals, and in adherence to the Guide for the Care and Use of Laboratory Animals, National Research Council, 1996.

## Rhesus macaque naïve B-cell repertoire sequencing

Frozen PBMCs from five naïve Indian origin RMs were obtained from Yerkes National Primate Research Center (IACUC approval YER2001036). The cells were rapidly thawed in a 37°C water bath and immediately diluted into 10 mL of pre-warmed RPMI media with 10% (v/v) heat-inactivated fetal bovine serum (FBS). Cells were pelleted at 400xg for 7 minutes and resuspended in 0.5 mL FACS buffer (PBS + 1% (v/v) FBS and stained on ice for 1 hr with the panel of fluorescent antibodies against IgM ([clone G20-127] BD), CD4 ([clone OKT-4] Bio-Legend), CD3 ([clone SP34-2] BD), IgG ([clone G18-145] BD), CD20 ([clone 2H7] BioLegend), CD8 ([clone RPA-T8] BioLegend), CD14 ([clone M5E2] BD), CD16 ([clone eBioCB16] ThermoFisher) and an eFluor780 viability marker (ThermoFisher). A MoFlo Astrios cell sorter (Beckman Coulter), with gating for live IgM$^+$/CD20$^+$/CD3$^-$/CD4$^-$/CD8$^-$/CD14$^-$/CD16$^-$ cells, was used to isolate cells that were then pelleted at 600xg for 10 min, resuspended in RLT+BME buffer (Qiagen), snap frozen in a dry ice ethanol bath and stored at -80°C. RNA extraction was performed using RNeasy Protect Mini kit (Qiagen) following the manufacturer's instructions. The 5′ rapid amplification of cDNA ends (5'RACE) with template switching method was used to obtain cDNA with unique molecular identifiers (UMIs) using a protocol modified from Turchaninova et al. [49]. Briefly, 300 ng of RNA was used in a 5'RACE cDNA synthesis reaction. The first-strand cDNA synthesis was performed at 42°C for 1 hr using the RM IgM outer reverse primer (5'-GTGATGGAGTCGGGGAAGGAAG-3'), a template switch adaptor with incorporated UMIs (5'-AAGCAGUGGUAUCAACGCAGAGUNNNNUNNNNUNNNNUC TTrGrGrGrG-3'), and SMARTScribe Reverse Transcriptase (Clontech). Residual template switch adaptor was removed by incubation with 5 U of uracil DNA glycosylase (New England BioLabs) for 40 min at 37°C. The resulting cDNA was purified using the MinElute PCR Purification Kit (Qiagen) following the manufacturer's instructions. PCR amplification was performed using the Q5 High-Fidelity DNA Polymerase (New England BioLabs), the forward primer (5'-NNNNAAGCAGTGGTATCAACGCA-3'), and the RM IgM inner reverse primer (5'-NNNNNAGGGGGAAAAGGGTTG-3'). Illumina adaptors were added using the NEB-Next Ultra II DNA Library Prep Kit (New England BioLabs) following the manufacturer's instructions. Libraries were sequenced on an Illumina MiSeq using the Illumina v3, (2x 300 bp) sequencing kit.

## Indian origin RM germline BCR database

Gene/alleles published by Vigdorovich et al. [31] were aligned to the Mmul_8.0.1 Indian origin RM genome assembly using BLAST [50,51]. Sequences that were not identical to the reference genome were eliminated. Additional full-length genes/alleles from available Indian origin RM genomic DNA sequencing datasets [21,52] were added to the database. Duplicates and sequences containing ambiguous bases were removed. The resulting initial Indian origin RM germline BCR database (S1 Table) was used for running IgDiscover on additional NGS datasets that were obtained during this study as described above, downloaded from the NCBI SRA

[27,30], or obtained directly from the study authors [31]. Paired sequence reads were aligned and filtered for length and quality using VDJServer [53]. Novel germline BCR gene/alleles were inferred using IgDiscover v0.11 with the germline_filter parameters "unique_cdr3s" and "unique_js" set to 10 and 4 respectively to reduce the rate of false positives [27]. Inferred genes/alleles were kept if they were detected in more than one animal or if they were identical to RM genes/alleles that were previously deposited in NCBI. The resulting germline BCR database (available at http://ward.scripps.edu/gld/) was converted into a custom IgBLAST database and subsequently used to analyze the BG505-specific mAb sequences [54].

## Env sequence analysis

Prevalence of specific amino acids or potential N-linked glycosylation sites (PNGS) were calculated using HIVAnchor (https://github.com/chazbot72/anchor). Pairwise alignments between the HxB2 Env reference sequence (K03455) and the LANL 2018 Group M super filtered web alignment was performed using Clustal Omega [55]. The results were parsed into a database keyed on positions relative to the reference, with gaps notated as sub positions following the last identical residue. The database was subsequently interrogated for conservation of amino acids or PNGS at specific positions.

## Env protein production

BG505 SOSIP.664, BG505 SOSIP.664-D7324 tag, BG505 SOSIP.664-AviTag, BG505 SOSIP. v4.1, and BG505 SOSIP.v5.2 were expressed in HEK293F cells and purified with either PGT145 or 2G12 affinity chromatograph followed by size exclusion chromatography (SEC) using a HiLoad 16/600 Superdex pg200 (GE Healthcare) as described previously [7,8,13]. Monomeric gp120 proteins (AviTag or D7324 tagged) were purified using a *Galanthus nivalis* lectin (Vector Labs) column. The Avi-tagged proteins were biotinylated using the BirA enzyme (Avidity) according to the manufacturer's protocol. The resulting biotinylated proteins are referred to using the descriptor AviB.

## Monoclonal antibody isolation

BG505 SOSIP.664-specific IgG[+] memory B-cells from isolated PBMCs from RMs rh1987 and rh2011 were single cell sorted in lysis buffer in order to amplify the antigen-specific mAbs, as previously described [56]. PBMCs were stained with primary fluorophore-conjugated antibodies to human CD3, CD8, CD14, CD20, IgG and IgM (BD Pharmigen). For staining with Env proteins, 50 nM of BG505 SOSIP.664-AviB, BG505 SOSIP.664 7C3-AviB or gp120-AviB were coupled in equimolar ratios to Streptavidin-PE, Streptavidin-FITC or Streptavidin-APC (Life Technologies), respectively. Cells were stained for 1 hr at 4°C in PBS supplemented with 1 mM EDTA and 1% FBS. In the gating strategy, we first excluded unwanted cell populations (CD3[-]/CD8[-]/CD14[-]) followed by selection of HIV Env-specific (positive for any of the 3 probes) memory B-cells (CD20[+]/IgG[+]/IgM[-]/HIV[+]). Cells of interest were single-cell sorted using a BD FACSAria III machine, into 96-well plates containing lysis buffer, and immediately stored at -80°C. One round of reverse-transcription and two rounds of nested PCR were performed to amplify the antibody V(D)J genes as previously described by Tiller *et al*. [15]. The PCR products containing the variable regions of the heavy chain or light chain, kappa or lambda were cloned into human IgG expression vectors to produce mAbs as described previously [56]. Fab expression vectors were made by introducing two stop codons following residue D234 (Kabat numbering [57]) in the IgG heavy chain vectors using the QuikChange Lightning Site-Directed Mutagenesis kit (Agilent). Sequences were verified by Sanger sequencing (Genewiz).

## Monoclonal antibody and Fab production

MAbs and Fabs were expressed in HEK293F cells and purified using affinity chromatography. Briefly, HEK293F cells (Invitrogen) were co-transfected with heavy and light chain plasmids (1:1 ratio) using PEImax. Transfections were performed according to the manufacturer's protocol. Supernatants were harvested 4–6 days following transfection and passed through a 0.45 μm filter. MAbs were purified using Protein A/G (ThermoFisher) or MAbSelect (GE Healthcare) affinity chromatography. Fabs were purified using CaptureSelect CH1-XL (ThermoFisher) affinity chromatography.

## D7324-capture ELISA for monomeric and trimeric BG505 Env proteins

Binding ELISAs were conducted as described previously [13,36].

## TZM-bl cell-based neutralization assays

Neutralization assays using the autologous BG505.T332N virus and mutants, and the heterologous SF162 virus, were carried out as described previously [58]. Nonlinear regression curves were determined and 50% inhibitory concentration ($IC_{50}$) values were calculated using a sigmoid function in Graphpad Prism v7.03.

## Bio-Layer Interferometry (BLI)

An Octet RED instrument (FortéBio) was used to determine the kinetic parameters of the antibody–antigen interactions by Biolayer Interferometry. Monoclonal Fabs were loaded onto anti-human Fab-CH1 (FAB2G) biosensors (FortéBio) at a concentration of 10 μg/mL in kinetics buffer (PBS, pH 7.4, 0.01% [w/v] BSA, and 0.002% [v/v] Tween 20) until a response of 1 nanometer shift was reached. Loaded biosensors were dipped into kinetics buffer for 1 min to acquire a baseline and then moved to wells containing a series of 2-fold dilutions of BG505 SOSIP.v5.2 in kinetics buffer, starting at a 4000 nM. The trimers were allowed to associate for 180 secs before the biosensor were move back to the wells containing kinetics buffer where the baseline was acquired. Disassociation of the trimers from the Fab-loaded biosensors was recorded for 300 secs. All BLI experiments were conducted at 37˚C. Kinetic parameters were calculated using the Octet System Data Analysis v9.0 (FortéBio).

## Negative Stain Electron Microscopy

BG505 SOSIP/Fab complexes were made by mixing 10–15 μg SOSIP with a 3 to 6-fold per protomer molar excess for monoclonal Fabs or 500 μg polyclonal Fabs and allowed to incubate for 18 to 24 hrs at room temperature (RT). Complex samples were either diluted to 0.02 mg/mL and applied to glow discharged negative stain grids or they were SEC purified using a Superose 6 Increase 10/300 GL (GE Healthcare) column to remove excess Fab prior to EM grid preparation. Fractions containing the SOSIP/Fab complexes were pooled and concentrated using 10 kDa Amicon spin concentrators (Millipore). Samples were diluted to 0.03 mg/mL in TBS (0.05 M Tris pH 7.4, 0.15 M NaCl) and adsorbed onto glow discharged carbon-coated Cu400 EM grids (Electron Microscopy Sciences) and blotted after 10 seconds. The grids were then stained with 3 μL of 2% (w/v) uranyl formate, immediately blotted, and stained again for 45 secs followed by a final blot. Image collection and data processing was performed as described previously on either an FEI Tecnai T12 microscope (2.05 Å/pixel; 52,000× magnification) or FEI Talos microscope (1.98 Å/pixel; 72,000× magnification) with an electron dose of $\sim 25$ electrons/$\text{Å}^2$ using Leginon [59,60]. 2D classification, 3D sorting and 3D refinement conducted

using Relion v3.0 [61]. EM density maps were visualized using UCSF Chimera and segmented using Segger [62,63].

## X-ray Crystallography Data Collection and Processing

All crystals were grown using sitting drop vapor diffusion. The RM20F Fab was crystallized from a solution containing 10 mg/mL protein in TBS with a well solution containing 0.1M MES, pH 5.0 and 2M ammonium sulfate. The crystals were cryoprotected by soaking in a well solution supplemented with 30% ethylene glycol. The RM20J Fab was crystallized from a solution containing 10 mg/mL protein in TBS with a well solution containing 0.1M MES, pH 6.0, 5% PEG3000 and 40% PEG400, with no cryoprotectant supplemented. The RM20E1 Fab was crystallized from a solution containing 6.3 mg/mL protein in TBS with a well solution containing 0.1M glycine, pH 10.5, 1.2M $NaH_2PO_4$, 0.8M $Na_2HPO_4$, and 0.2M $Li_2SO_4$, with 15% ethylene glycol supplemented as cryoprotectant. All crystals were grown at 298 K. Diffraction data for RM20F and RM20E1 were collected at the Stanford Synchrotron Radiation Lightsource (SSRL) beamline BL12–2, and that for RM20J collected at the Advanced Photon Source (APS) beamline 23ID-B. Data collection and processing statistics are detailed in S5 Table. Data sets were indexed, integrated, and scaled using the HKL-2000 package [64]. The structures were solved by molecular replacement using PHASER [65] with a homology model (SWISS-MODEL; [66–68]) as a search model and further refined using phenix.refine [69] combined with manual building cycles in Coot [70].

## Cryo Electron Microscopy Sample Preparation

RM20J complex: 500 μg BG505 SOSIP.v5.2 was mixed with 656 μg RM20J Fab and incubated at RT overnight. The complex was SEC purified using a HiLoad 16/600 Superdex pg200 (GE Healthcare) column in TBS. Fractions containing the complex were concentrated to 6.1 mg/mL using a 10 kDa Amicon spin concentrator (Millipore). 3.5 μL of the complex was mixed with 0.57 μL of 0.04 mM lauryl maltose neopentyl glycol (LMNG) and applied to a C-Flat grid (CF-2/1-4C, Protochips, Inc.), which had been plasma-cleaned for 5 seconds using a mixture of $N_2/O_2$ (Gatan Solarus 950 Plasma system). The grid was blotted and plunged into liquid ethane using a Vitrobot Mark IV (ThermoFisher).

RM20F complex: 500 μg BG505 SOSIP.v4.1 was mixed with approximately 1,000 μg RM20F Fab and incubated at RT overnight. The complex was SEC purified using a Superose 6 Increase 10/300 GL (GE Healthcare) column in TBS. Fractions containing the complex were concentrated to 6 mg/mL using a 10 kDa Amicon spin concentrator (Millipore). 3 μL of the complex was mixed with 1 μL of a n-Dodecyl-β-D-Maltopyranoside (DDM) solution to a final DDM concentration of 0.06 mM and applied to a C-Flat grid (CF-2/2-4C, Protochips, Inc.), which had been plasma-cleaned for 5 seconds using a mixture of $N_2/O_2$ (Gatan Solarus 950 Plasma system). The grid was blotted and plunged into liquid Ethane using a Vitrobot Mark IV (ThermoFisher).

RM20E1 complex: 355 μg BG505 SOSIP.v5.2 was mixed with 484 μg RM20E1 Fab and 484 μg PGT122 Fab and incubated at RT overnight. The complex was SEC purified using a HiLoad 16/600 Superdex pg200 (GE Healthcare) column in TBS. Fractions containing the complex were concentrated to 4 mg/mL using a 10 kDa Amicon spin concentrator (Millipore). 3 μL of the complex was mixed with 1 μL of a n-Dodecyl-β-D-Maltopyranoside (DDM) solution to a final DDM concentration of 0.06 mM and applied to a grid (Quantifoil R 1.2/1.3, 400), which had been plasma-cleaned for 5 seconds using a mixture of $N_2/O_2$ (Gatan Solarus 950 Plasma system). The grid was blotted and plunged into liquid Ethane using a Vitrobot Mark IV (ThermoFisher).

## Cryo Electron Microscopy Data Collection and Processing

Samples were imaged on either FEI Titan Krios electron microscope (ThermoFisher) operating at 300 keV (RM20F dataset) or a FEI Talos Arctica electron microscope (ThermoFisher) operating at 200 keV (RM20J and RM20E1 datasets). Both microscopes were equipped with Gatan K2 Summit direct electron directors operating in counting mode. Automated data collection was performed using the Leginon software suite [59]. Micrograph movie frames were aligned and dose-weighted using MotionCor2 [71], and CTF models were determined using Gctf [72]. Particle picking, 2D classification, Ab-initio reconstruction, and 3D refinement were conducted using cryoSPARCv2 [73]. Data collection and processing parameters are reported in S4 Table.

Initial molecular models of the BG505 SOSIP trimer/Fab complexes were built by docking the Env portion of PDB: 5V8M [74] into the EM density maps along with the relevant Fab crystal structures (PDB: 4JY5 was used for PGT122 [75]) using UCSF Chimera [62]. The Fab constant regions were removed due to flexibility in the elbow region as commonly found in Fab structures [76], the appropriate stabilizing mutations (v4.1 or v5.2) were introduced into the Env sequence, and N-linked glycans were added using Coot [77]. The models were iteratively refined into the EM density maps using RosettaRelax and Coot [70,77–80]. Glycan structures were validated using Privateer [81]. Overall structures were evaluated using EMRinger [82] and MolProbity [83]. Protein interface calculations were performed using jsPISA [84]. Final model statistics are summarized in S4 Table.

## Statistical analysis

Statistical models inherent to Relion 3.0 [61] and cryoSPARC [73] were employed in image analysis to derive 2D classes and 3D models. All ELISA and neutralization assays were conducted with at least duplicate measurements.

## Supporting information

**S1 Fig. MAb isolation and characterization from BG505 SOSIP.664 trimer-immunized macaques.** (A) FACS gating strategy for isolation of BG505 SOSIP specific memory B-cells. (B) BG505 SOSIP.664 trimer and BG505 gp120 ELISA binding data for mAbs isolated from RM rh1987 and (C) from RM rh2011.
(PDF)

**S2 Fig. Negative stain electron microscopy epitope mapping of Fabs from rh1987 and rh2011.** Representative 2D class averages, 3D reconstructions, and EMDB accession numbers.
(PDF)

**S3 Fig. Cryo-EM structures of BG505 SOSIP trimers with mAbs and crystal structures of mAbs.** (A) Representative micrographs for cryoEM datasets. (B) Local resolution maps for each complex generated in cryoSPARC v2 (Punjani *et al.*, 2017). (C) Gold-standard Fourier shell correlation (FSC) curves for each complex showing global resolution calculated at FSC = 0.143. (D) Crystal structures of unliganded Fabs RM20J, RM20F, and RM20E1. Heavy chains are shown in green and light chains shown in light blue.
(PDF)

**S4 Fig. Antibody bound FP conformations.** The HIV Env trimer is shown as a surface representation with one protomer colored in grey, the other two protomers colored in blue, and the N-linked glycans colored in green. The FP (residues 512–522) from the antibody bound structures of RM20E1, RM20F, DFPH-a.15 (PDB: 6N1W), VRC34 (PDB: 5I8H), and ACS202

(PDB: 6NC2) are shown as colored backbone ribbon diagrams.
(PDF)

**S1 Table. Initial germline BCR database used in IgDiscover.**
(XLSX)

**S2 Table. MAb CharacteristicsTables.**
(PDF)

**S3 Table. MAb neutralization, ELISA binding, and EM epitope mapping.**
(PDF)

**S4 Table. EM Data Collection and Map/Model Refinement Parameters.**
(PDF)

**S5 Table. X-ray data collection and refinement statistics.**
(PDF)

**S6 Table. BLI Binding Kinetics.**
(PDF)

**S7 Table. Heterologous neutralization.**
(PDF)

## Acknowledgments

We thank Jean-Christophe Ducom, Hannah Turner, and Bill Anderson for assistance with computational resources, microscope management, and data collection. We thank Alan Saluk, Brian Seegers, Steven Head, Jessica Ledesma, Curt Wittenberg, and Nitya Bhaskaran for assistance with naïve B-cell sorting and NGS sequencing.

## Author Contributions

**Conceptualization:** Christopher A. Cottrell, Rogier W. Sanders, Andrew B. Ward, Marit J. van Gils.

**Data curation:** Christopher A. Cottrell, Jelle van Schooten, Meng Yuan, David Oyen.

**Formal analysis:** Jelle van Schooten, Meng Yuan, David Oyen, Marit J. van Gils.

**Funding acquisition:** Dennis R. Burton, John P. Moore, Ian A. Wilson, Rogier W. Sanders, Andrew B. Ward, Marit J. van Gils.

**Investigation:** Christopher A. Cottrell, Jelle van Schooten, Meng Yuan, David Oyen, Mia Shin, Robert Morpurgo, Patricia van der Woude, Mariëlle van Breemen, Jonathan L. Torres, Raj Patel, Leigh M. Sewall, Jeffrey Copps, Gabriel Ozorowski, Bartek Nogal, Devin Sok, Celia Labranche, Marit J. van Gils.

**Methodology:** Christopher A. Cottrell, Jelle van Schooten, Meng Yuan, David Oyen, Marit J. van Gils.

**Project administration:** Ian A. Wilson, Rogier W. Sanders, Andrew B. Ward, Marit J. van Gils.

**Resources:** Eva G. Rakasz, Vladimir Vigdorovich, Diane G. Carnathan, D. Noah Sather, David Montefiori, Guido Silvestri, Dennis R. Burton, John P. Moore.

**Software:** Charles A. Bowman, Justin Gross, Scott Christley.

**Supervision:** Ian A. Wilson, Rogier W. Sanders, Andrew B. Ward, Marit J. van Gils.

**Validation:** Christopher A. Cottrell, Marit J. van Gils.

**Visualization:** Christopher A. Cottrell, Jelle van Schooten.

**Writing – original draft:** Christopher A. Cottrell, Marit J. van Gils.

**Writing – review & editing:** Christopher A. Cottrell, Jelle van Schooten, Dennis R. Burton, John P. Moore, Ian A. Wilson, Rogier W. Sanders, Andrew B. Ward, Marit J. van Gils.

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
