## [Decision Letter · Decision Letter 0]

19 May 2020

Dear Dr. van Gils,

Thank you very much for submitting your manuscript "Mapping the immunogenic landscape of near-native HIV-1 envelope trimers in non-human primates" for consideration at PLOS Pathogens. As with all papers reviewed by the journal, your manuscript was reviewed by members of the editorial board and by several independent reviewers. The reviewers appreciated the attention to an important topic. Based on the reviews, we are likely to accept this manuscript for publication, providing that you modify the manuscript according to the review recommendations.

Sincerely,

Ronald C. Desrosiers

Associate Editor

PLOS Pathogens

Thomas Hope

Section Editor

PLOS Pathogens

Kasturi Haldar

Editor-in-Chief

PLOS Pathogens

orcid.org/0000-0001-5065-158X

Michael Malim

Editor-in-Chief

PLOS Pathogens

orcid.org/0000-0002-7699-2064

Reviewer Comments (if any, and for reference):

Reviewer's Responses to Questions

**Part I - Summary**

Reviewer #1: In this manuscript, Cottrell et al. report the characterization of 42 new monoclonal antibodies isolated from two rhesus macaques previously immunized with the BG505 SOSIP.664 trimer. All of the antibodies bound the aforementioned trimer. Six of them neutralized the autologous pseudovirus BG505.T332N (with modest potency) and 9 neutralized a N611-glycan deficient variant BG505.T332.N611A (some with high potency). Out of the neutralizing antibodies, three neutralized both viruses. Extensive characterization of all 42 antibodies (neutralizing and non-neutralizing) revealed their antigenic specificities: 55% of the 42 antibodies bound the base of the trimer, 19% the 289-glycan hole, 12% an epitope near the N611 glycan, and 2% the gp120/gp41 interphase. The antibody with the most potent neutralizing activity against BG505.T332N bound the gp120/gp41 interphase in an epitope that overlaps with VRC34.01, a previously identified broadly neutralizing antibody of human origin.

In my opinion the data is clear, and the manuscript is well written and balanced. Remarkably, the authors did a big effort in putting together a comprehensive dataset of rhesus germline sequences to be used with their analyses. In fact, having a good database of sequences is important for proper identification of the corresponding germline genes and alleles, and this translates into accurate SHM rate calculations. Current databases are indeed limited, at least in regard to rhesus sequences. Importantly, the authors generated an online tool for antibody sequence analysis with their newly generated database: http://ward.scripps.edu/gld/. The tool/site was made freely available and will certainly be useful for such analyses. Hopefully more sequences will be added over time to keep it growing.

The antibody epitope analyses were very thorough, and I agree with the authors that extending those to all the antibodies rather than focusing only on the neutralizing ones was a good idea, with the ultimate goal of improving future designs of SOSIP trimers.

Reviewer #2: The manuscript by Cottrell et al. describes a series of in depth analyses of monoclonal antibodies generated from two rhesus macaques that were immunized using the HIV-1 Env BG505 SOSIP.664 trimer. These two macaques were chosen because they showed higher levels of neutralizing antibody after immunization. Similar experiments have been done in rabbits, and will soon be done with samples from humans being immunized with this immunogen. A comparative study is of value in assessing the predictive value of model systems given the small number of immunogens that will ever be tested in humans. The study is comprehensive in that it goes from the immunized animals all the way to the structure of mAbs bound to the trimer. This represents a serious attempt to characterize the nature of the antibodies raised in macaques by the SOSIP trimer. Even if it ultimately does not inform the nature of the immune response in humans it still represents an important data point on the path to vaccine development. In addition, the authors expand the number of rhesus macaque germline BCR alleles to help interpret somatic hypermutations. I have only a few minor comments.

Reviewer #3: In “Mapping the immunogenic landscape of near-native HIV-1 envelope trimers in non-human primates,” Cottrell et al. describe the isolation of 42 monoclonal antibodies from two Indian origin macaques, immunized repeatedly over a year with the BG505 SOSIP.664 immunogen. Sixteen of these antibodies were characterized for recognition by negative stain EM, to provide a general notion of epitope – then further characterized at higher resolution for three of the antibodies: RM20J, which recognized a glycan hole at residue 289; RM20F, which recognized an interface epitope; and RM20E1, which recognized a quaternary epitope including the fusion peptide. While only 6 of the 42 antibodies were capable of neutralizing the autologous BG505 virus, the authors argue – I believe correctly – that collectively, these antibodies delineate many of the most immunogenic features of the SOSIP-based immunogen. These antibodies and their mapping thus represent a helpful resource, delineating the dominant immunogenic characteristics of an important HIV-1 immunogen that has entered clinical trials.

**Part II – Major Issues: Key Experiments Required for Acceptance**

Reviewer #1: (No Response)

Reviewer #2: None

Reviewer #3: While okay to focus on strain-strain specific and non-neutralizing antibodies, to provide perspective for their results, the authors should discuss/compare their antibodies with the few broadly neutralizing antibodies, which have now been elicited from macaques with modified SOSIP trimers (e.g. Dubrovskaya et al. 2019, Immunity and Kong et al. 2019, Cell). The authors cite these papers when they state, “Native-like Env trimers are an important design platform for engineering immunogens for bNAb induction.” (Lines 274-275), but they provide little comparison/context. They should compare their interface antibody (RM20F) with the interface antibody (1C2 with 87% breadth) from the Dubrovskaya manuscript and their fusion peptide antibodies (including RM20E1) with the fusion peptide antibody (DFPH-a.01 with 59% breadth) from the Kong manuscript. What are key differences between the broadly neutralizing antibodies versus the strain-specific or non-neutralizing ones characterized in their study? (SHM? Angle of approach?) While fine to focus on the diverse landscape of immunodominant antibodies, it’s important to provide context as to where broadly neutralizing antibodies elicited in macaques lie within that landscape.

**Part III – Minor Issues: Editorial and Data Presentation Modifications**

Reviewer #1: I only have some minor comments/questions:

- The neutralizing antibodies described here targeted varied epitopes. For instance, RM19M (which neutralized both BG505.T332N and BG505.T332N.N611A) bound the base of the SOSIP, but others bound the glycan hole, etc. For enhanced clarity, I suggest including a supplementary table in which the info from the tables in Fig 1C and D is added and combined (adding extra columns) with the ELISA reactivity to SOSIP and gp120, as well as the information on the corresponding epitope for each antibody.

- A total of 42 antibodies were isolated (and 23 out of those bound the base of the trimer). Would the 19 remaining antibodies be representative or enough to cover all of the non-neutralizing epitopes found on the SOSIP trimer?

- What was the criteria used to select the 40 viruses used to test neutralization of RM20F and VRC34 in Figure S4?

- If authors were trying to obtain as many sequences as possible for their database, why were those that were not identical to the one reference genome eliminated? In line with this, how many different animals are represented in the germline sequence database?

Reviewer #2: 1. The authors pick out two immunized macaques to analyze form mAbs. While they state these were the best two (out of four?) for neutralizing titers it is hard for the reader to see the data that is the basis for this choice. Perhaps it is in their earlier paper from 2015, which seems to be about immunized rabbits for the most part. Including a figure or table showing the neutralization titer and Tier 2 breadth of the starting serum from either these two or even all of the animals that were in the study where these two were included would be useful. This is important because there will be multiple studies like this and it will be informative to understand as much about the animal immune response to allow comparison between studies.

2. It would be useful to include in the Discussion a comparison of antibodies in rabbits, macaques, and humans (and maybe guinea pigs). As one example, who can make long CDR3 loops and how often is this an important feature of a neutralizing antibody? More generally, what are the important features of neutralizing antibodies and how well do these different species generate antibodies with the relevant features? This would help the reader frame how to compare results in different species, ultimately to help understand which will be most informative in modeling humans.

Reviewer #3: In terms of minor issues with presentation: the figures are generally clear, and well labeled, though yellow fonts should be darkened, and some of the writing should be improved.

The first two sections of the results should be edited to move methodological (what was done) details to Methods; this first part of the results paper makes the case for accurate SHM determination, though I don’t see SHM information highlighted in the description of structures – only in the penultimate paragraph of the paper.

Information on lines 123-147 seems better suited to a table.

PLOS authors have the option to publish the peer review history of their article (what does this mean?). If published, this will include your full peer review and any attached files.

Reviewer #1: Yes: Jose Martinez-Navio

Reviewer #2: No

Reviewer #3: No
---

## [Editor Report · Decision Letter 1]

26 Jun 2020

Dear Dr. van Gils,

We are pleased to inform you that your manuscript 'Mapping the immunogenic landscape of near-native HIV-1 envelope trimers in non-human primates' has been provisionally accepted for publication in PLOS Pathogens.

Best regards,

Ronald C. Desrosiers

Associate Editor

PLOS Pathogens

Thomas Hope

Section Editor

PLOS Pathogens

Kasturi Haldar

Editor-in-Chief

PLOS Pathogens

orcid.org/0000-0001-5065-158X

Michael Malim

Editor-in-Chief

PLOS Pathogens

orcid.org/0000-0002-7699-2064
---

## [Editor Report · Acceptance letter]

19 Aug 2020

Dear Dr. van Gils,

We are delighted to inform you that your manuscript, "Mapping the immunogenic landscape of near-native HIV-1 envelope trimers in non-human primates," has been formally accepted for publication in PLOS Pathogens.

Best regards,

Kasturi Haldar

Editor-in-Chief

PLOS Pathogens

orcid.org/0000-0001-5065-158X

Michael Malim

Editor-in-Chief

PLOS Pathogens

orcid.org/0000-0002-7699-2064